# Wildlife vaccination strategies for eliminating bovine tuberculosis in white-tailed deer populations

**Aakash Pandey** [1] *, **Abigail B. Feuka** [2], **Melinda Cosgrove** [3], **Megan Moriarty** [3], **Anthony Duffiney** [4], **Kurt C. VerCauteren** [2], **Henry Campa, III** [1], **Kim M. Pepin** [2]

**1** Department of Fisheries and Wildlife, Michigan State University, East Lansing, Michigan, United States of America, **2** National Wildlife Research Center, Wildlife Services, Animal and Plant Health Inspection Service, United States Department of Agriculture, Fort Collins, Colorado, United States of America, **3** Wildlife Disease Laboratory, Wildlife Division, Michigan Department of Natural Resources, Lansing, Michigan, United States of America, **4** Wildlife Services, Animal and Plant Health Inspection Service, United States Department of Agriculture, Okemos, Michigan, United States of America

* pandeyaa@msu.edu

**Data Availability Statement:** Data used in this study is provided in the supplemental information (S1-S4 Tables). Simulation codes are available at 10.5281/zenodo.8060966.

## Abstract

Many pathogens of humans and livestock also infect wildlife that can act as a reservoir and challenge disease control or elimination. Efficient and effective prioritization of research and management actions requires an understanding of the potential for new tools to improve elimination probability with feasible deployment strategies that can be implemented at scale. Wildlife vaccination is gaining interest as a tool for managing several wildlife diseases. To evaluate the effect of vaccinating white-tailed deer (*Odocoileus virginianus*), in combination with harvest, in reducing and eliminating bovine tuberculosis from deer populations in Michigan, we developed a mechanistic age-structured disease transmission model for bovine tuberculosis with integrated disease management. We evaluated the impact of pulse vaccination across a range of vaccine properties. Pulse vaccination was effective for reducing disease prevalence rapidly with even low (30%) to moderate (60%) vaccine coverage of the susceptible and exposed deer population and was further improved when combined with increased harvest. The impact of increased harvest depended on the relative strength of transmission modes, i.e., direct vs indirect transmission. Vaccine coverage and efficacy were the most important vaccine properties for reducing and eliminating disease from the local population. By fitting the model to the core endemic area of bovine tuberculosis in Michigan, USA, we identified feasible integrated management strategies involving vaccination and increased harvest that reduced disease prevalence in free-ranging deer. Few scenarios led to disease elimination due to the chronic nature of bovine tuberculosis. A long-term commitment to regular vaccination campaigns, and further research on increasing vaccines efficacy and uptake rate in free-ranging deer are important for disease management.

**Funding:** This work was supported by the US Federal Aid in Wildlife Restoration Act under Michigan Pittman-Robertson Project W-147-R. The data used in this study were provided by the funding agency. MC and MM are affiliated with the funding agency and provided input on the manuscript text for a management context.

**Competing interests:** The authors have declared that no competing interests exist.

## Author summary

Many wildlife species act as reservoirs of bovine tuberculosis from which they can transmit to cattle and humans. In Michigan, bovine tuberculosis is endemic in white-tailed deer causing occasional spillovers to cattle. Here, with the help of a disease transmission model, we explored the use of wildlife vaccination in white-tailed deer to manage bovine tuberculosis in Michigan. We show that pulse vaccination effectively reduces disease prevalence, especially when combined with increased deer harvest. Key vaccine parameters for disease elimination are vaccine coverage and efficacy. While complete elimination is challenging, sustained vaccination and research to improve vaccine effectiveness are crucial for long-term disease management. Hence, this research informs integrated strategies to reduce bovine tuberculosis in deer populations.

## Introduction

Pathogens with wildlife reservoirs present a challenge for disease management because wildlife populations range freely and their eco-epidemiological processes are often poorly understood [1]. This collective challenge makes it difficult to predict the effects of different management techniques. *Mycobacterium bovis* (*M. bovis*), the causative agent of bovine tuberculosis (bTB), infects various wildlife species, which can transmit the pathogen to cattle and humans. Due to the broad host-range of *M. bovis*, different wildlife species act as a reservoir in various locations [2]. White-tailed deer (*Odocoileus virginianus*), elk (*Cervus canadensis*), and bison (*Bison bison*) are believed to be major reservoirs in North America [2,3], whereas European badger (*Meles meles*) is thought to be the dominant reservoir in England and Ireland [4]. Wild boar (*Sus scrofa*), brush tailed possum (*Trichosurus vulpecula*) and African buffalo (*Syncerus caffer*) are considered bTB reservoirs in Spain, New Zealand, and South Africa, respectively [2]. A good understanding of system-specific disease dynamics in wildlife populations and management practices grounded in disease ecology theory are required for effective management of wildlife diseases [1].

Infectious diseases in wildlife can be controlled by reducing the susceptible population available for infection. This can be achieved by: I) reducing the overall host population density through techniques such as random or selective culling or harvest, reproductive control, and/ or habitat management [5], and II) reducing the susceptible population through immunization [6–8]. Culling and harvest have been used to control diseases in wildlife for decades, including bTB in wildlife reservoirs [8]. For example, random culling of European badger in the UK and Ireland [9–14], brush-tailed possum in New Zealand [15], wild boar in Spain [16,17], and feral water buffalo in Australia [18] were performed to reduce host population size to control bTB prevalence. In the USA, random and selective harvest have been used in conjunction with implementing feeding and baiting bans in attempt to control bTB in white-tailed deer populations [19–21]. Increased harvest, however, may not always be feasible due to ethical, socio-political and economic concerns, and lack of public support to harvest additional animals [21,22]. Vaccination by oral baiting has been an effective strategy in reducing, and in some cases eliminating, rabies in several free-ranging species [23–25]. In the case of bTB, lab and field experiments with existing bacille Calmette-Guérin (BCG) vaccines have been shown to offer protection in brush-tailed possum [26,27], European badger [28,29], wild boar [30,31]. Oral delivery of BCG vaccine in white-tailed deer has produced similar results in lab trials where vaccinated deer were less likely to develop gross lesions after experimental challenge

with virulent *M. bovis* compared to unvaccinated deer [32–35]. Vaccination is thus proposed as an alternative strategy to control bTB in deer [32–36].

The development and use of vaccines for wildlife is expensive and time consuming [37]. Mathematical models of disease dynamics with interventions can be used to evaluate a range of conditions that would otherwise be infeasible through field-based experimentation to identify effective management strategies for specific ecological conditions and vaccine properties. Previous modeling research has shown that vaccination can be used as a tool in bTB elimination in white-tailed deer, but requires a long-term effort [38,39]. However, studies focusing on the deployment regimes and vaccine properties needed for elimination are lacking. Also, simulation models provide the most realistic results when parameters are estimated from a system's surveillance data, a gap that exists in previous studies. In this study, we focus on imperfect vaccines with a wide range of vaccine coverages, or proportions of the population vaccinated, in susceptible and exposed populations. An imperfect vaccine can be leaky when it reduces, but does not eliminate, the potential for infection [40]. Similarly, vaccines can affect disease manifestation leading to differential transmission rates from vaccinated and unvaccinated infected individuals. Lastly, imperfect vaccines do not provide complete protection forever, and exploring the effect of different immunity waning rates can help inform how often vaccination needs to occur. We developed and discussed a mechanistic disease transmission model that we fit to bTB surveillance data in white-tailed deer from the four-county core area of bTB in Michigan, USA. We used the fitted models to identify (I) integrated white-tailed deer management strategies that include harvest and vaccination, and (II) vaccine properties including coverage, leakiness, relative transmissibility of vaccinated individual to that of unvaccinated individual, and immunity waning rate that dramatically reduce or eliminate bTB from free-ranging deer in these areas. Our results are important for designing effective vaccines for endemic diseases that cause chronic infection, such as bTB, and planning vaccination delivery strategies that will be effective for reaching management objectives.

## Methods

### Study area and data

In Michigan, bTB is endemic in deer populations in Alpena, Alcona, Montmorency, Oscoda, and Presque Isle counties. The cases are primarily localized in deer management unit (DMU) 452 (~1480 km$^2$) that occurs at the intersection of Alpena, Alcona, Montmorency and Oscoda counties [41]. Outbreaks of the disease in cattle have been linked to direct and indirect contact between infected deer and domestic cattle and to the movement of infected cattle among farms [41–43]. The Michigan Department of Agriculture and Rural Development (MDARD) and the Michigan Department of Natural Resources (MDNR) have implemented strict testing, surveillance and management programs, including mandatory annual whole herd and movement testing of cattle for bTB, farm biosecurity measures such as fencing and strict movement controls, voluntary bTB testing of hunter-harvested deer, bTB testing of targeted removal of habituated deer around cattle facilities, and a ban on baiting and feeding [41]. Although these practices have significantly reduced bTB prevalence in deer in recent decades [44] and consequently lowered spillback events in cattle, bTB has not been eliminated suggesting these interventions may have reached their capacity in reducing bTB prevalence in deer [41].

We used the data collected by MDNR from the four counties associated with DMU 452 from years 2011 to 2021 (Alpena, Alcona, Montmorency, and Oscoda). The procedure for data collection was described previously [45]. Briefly, deer samples were obtained from three major sources: voluntary submission from deer hunters, "non-hunt" deer including road kills and deer removed by landowners, and United States Department of Agriculture Wildlife

Services via disease control permits. Much of the data (~81%) came from hunter harvested deer. The location of kill or harvest was documented, and the age of individual deer was estimated using tooth wear patterns as previously described [45]. Deer heads were visually examined for the presence of gross lesions or abscessation. TB was considered not detected if no gross lesions were noted. Samples with gross lesions were further tested using histopathology, staining for acid-fast bacteria and confirmed via mycobacterial culture. Samples from which a positive culture of *M. bovis* was obtained were considered bTB positive. We used countywide apparent prevalence (percentage of positive samples out of all samples submitted for examination and testing) data to fit to the model. The prevalence data were categorized into three age classes: fawn (<1 year old), yearling ($\geq$ 1 year and <2 years old), and adult ($\geq$ 2 years old) and fitted to the age-structured disease model shown below.

## Disease model

We consider a deterministic age-structured SEI (Susceptible-Exposed-Infected) disease model (Fig 1) with pulse vaccination. The deer population was structured into fawns (F) (0–11 months old), yearlings (Y) (12–23 months old), and adults (A) (>23 months old). S, E, and I with F, Y, and A subscripts represent susceptible, exposed and infected numbers for each age class. Fawns were recruited via birth from yearling and adult females at rates $b_Y(t)$ and $b_A(t)$, respectively. Given that deer only give birth in late spring/early summer, $b_Y$ and $b_A$ are pulse birth rates where birth rates were assumed to be 0 except for a month ($t$) during which the birth occurs, i.e.,

$$b_Y(t) = \begin{cases} b_{Y0} & mod(t,12) = 0 \\ 0 & mod(t,12) \neq 0 \end{cases}$$

$$b_A(t) = \begin{cases} b_{A0} & mod(t,12) = 0 \\ 0 & mod(t,12) \neq 0 \end{cases}.$$

We assumed 1:1 sex ratio of female to male for each age class where the number of female fawns, yearlings, and adults were given as $N_F$, $N_Y$, and $N_A$. We further assumed a density-dependent birth rate given by a logistic equation with carrying capacity K. Deer were exposed to the pathogen either via a direct route at rate $\beta$, or via an environmental route at rate $\gamma$. We assumed a mass action function for direct transmission whereas the environmental transmission was modeled as a saturating function. Although the exact form of environmental transmission is not known, this type of functional form is used with other environmental pathogens, such as *Vibrio cholerae* [46,47]. Environmental transmission is considered an important infection source for bTB in wildlife including deer [48]. Exposed individuals get infected at rates $\epsilon_F$, $\epsilon_Y$, and $\epsilon_A$ in fawn, yearling and adult age classes. Thus, $1/\epsilon_F$, $1/\epsilon_Y$, and $1/\epsilon_A$ represent the age-specific latent periods. The rates of aging from fawn to yearling and yearling to adult are $y$ and $a$, respectively. Fawns, yearlings, and adults have age-specific hunter harvest rates ($h_F$, $h_Y$, $h_A$) and deaths due to other causes including natural causes of mortality ($d_F$, $d_Y$, $d_A$). For simplification, we assumed continuous harvest of deer throughout the year rather than harvest concentrated during the hunting season. This is a reasonable assumption given that the USDA removed deer outside of the regular hunting seasons. To test whether harvesting deer during a short period (e.g., hunting season) had any effect on disease dynamics, we performed sensitivity analysis with a model where harvest is assumed to be a pulse rather than continuous throughout a year. We assumed that bTB does not cause any measurable death in deer [38] but once deer are infected, they remain in the infected class for life. Infected deer

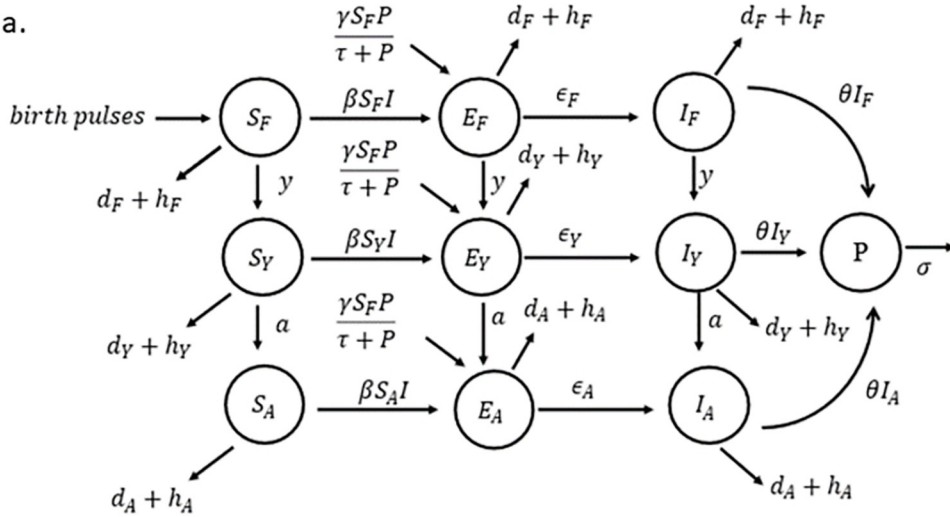

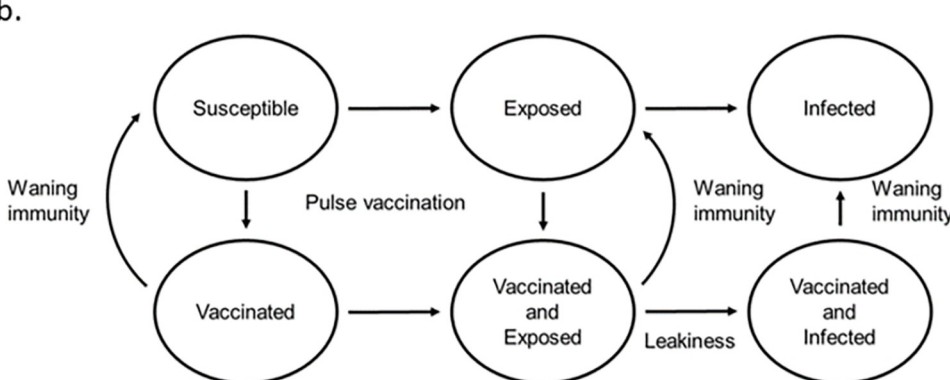

**Fig 1.** a. Flow diagram for age structured bTB disease model in white-tailed deer. Deer population is divided into fawn (F), yearling (Y), and adult (A). Each age class has susceptible (S), exposed (E), and infected (I) classes with subscripts F, Y, and A representing each age class. P represents number of live propagules in environment. See Table 1 for parameter definitions. b. Pulse vaccination with leaky vaccines. Susceptible and exposed individuals can be vaccinated.

shed the pathogen in the environment at rate $\theta$ and the pathogen survives on average for $1/\sigma$ months in the environment. The number of live pathogens in the environment is represented by P.

We assume pulse vaccinations at times $t_V$. At each vaccination, $p$ proportions of susceptible and exposed yet uninfected individuals in each class of yearling and adult from the preceding time point ($t_V^-$) gets vaccinated, i.e.,

$$v(t) = \begin{cases} p & t = t_V \\ 0 & t \neq t_V \end{cases}$$

The numbers of vaccinated yearling and adult are represented by $V_Y$ and $V_A$ respectively. The proposed primary (and most feasible) mode of vaccine distribution to wildlife is through oral vaccine delivery. Previous research in Michigan has shown that distributing placebo vaccines in baits with rhodamine B as a biomarker during late winter/early spring when food is

scarce, led to 69% of sampled deer (n = 107) being marked [49]. Hence, we model applying vaccination during this timeframe. As annual birth pulses occur just after this period, we assume fawns do not get vaccinated. Previous research on BCG vaccine have shown that it does not offer complete protection but rather reduces clinical signs. We thus consider an imperfect vaccine that reduces but does not eliminate the potential for infection. This is also known as the leakiness of the vaccine. More specifically, we assume that vaccinated individuals get exposed to *M. bovis* pathogens at the same rate as unvaccinated but the transition from the exposed to infected in vaccinated individuals is reduced by $\epsilon_l$ compared to unvaccinated individuals. We denote the $\epsilon_l$ as the leakiness parameter of the vaccine. Vaccine efficacy, a measurement of the protection from vaccine by the reduction in infection risk in vaccinated individual relative to that of unvaccinated individual, is then given as $1-\epsilon_l$. Although the vaccine may not provide perfect protection, it can alter disease manifestation leading to differential disease transmission rates from infected individuals who are vaccinated vs unvaccinated individuals. The relative transmissibility, i.e., the ratio of disease transmission rate from a vaccinated but infected individual to the disease transmission rate from an unvaccinated but infected individual is assumed to be $\kappa$. Shedding of the pathogen from the vaccinated but infected class is also assumed to be reduced by the same factor. We further assume that if an individual is already in the infected class, they do not receive any benefits of vaccination. Hence, we assume that although vaccinated individuals can get infected due to vaccine failures, already infected individuals cannot change into the vaccinated class. Exposed and infected individuals in vaccinated class are differentiated from exposed and infected individuals in unvaccinated class by the subscript *V*. We further assume that protection from the vaccine wanes at rate $\alpha$. Although the exact rate of immunity waning from the BCG vaccine is not known in deer, research in cattle shows that significant immune protection is present at 12, but not 24 months, post vaccination [50].

The following system of equations describes the disease dynamics with pulse vaccination:

Fawn Dynamics:

$$\frac{dS_F}{dt} = (b_Y(t)N_Y + b_a(t)N_A)\left(1 - \frac{2(N_F + N_Y + N_A)}{K}\right) - \beta S_F(I + \kappa(I_{VY} + I_{VA})) - \frac{\gamma S_F P}{\tau + P} - (d_F + h_F)S_F - yS_F$$

$$\frac{dE_F}{dt} = \beta S_F(I + \kappa(I_{VY} + I_{VA})) + \frac{\gamma S_F P}{\tau + P} - (d_F + h_F)E_F - \epsilon_F E_F - yE_F$$

$$\frac{dI_F}{dt} = \epsilon_F E_F - (d_F + h_F)I_F - yI_F$$

$$N_F = \frac{S_F + E_F + I_F}{2}$$

$$N_Y = \frac{S_Y + E_Y + E_{VY} + I_Y + I_{VY}}{2}$$

$$N_A = \frac{S_A + E_A + E_{VA} + I_A + I_{VA}}{2}$$

$$I = I_F + I_Y + I_A$$

Yearling Dynamics:

$$\frac{dS_Y}{dt} = yS_F - \beta S_Y(I + \kappa(I_{VY} + I_{VA})) - \frac{\gamma S_Y P}{\tau + P} - (d_Y + h_Y)S_Y - aS_Y + \alpha V_Y$$

$$\frac{dE_Y}{dt} = \beta S_Y(I + \kappa(I_{VY} + I_{VA})) + \frac{\gamma S_Y P}{\tau + P} + yE_F - (d_Y + h_Y)E_Y - aE_Y - \epsilon_Y E_Y + \alpha E_{VY}$$

$$\frac{dE_{VY}}{dt} = \beta V_Y(I + \kappa(I_{VY} + I_{VA})) + \frac{\gamma V_Y P}{\tau + P} - (d_Y + h_Y)E_{VY} - aE_{VY} - \epsilon_Y \epsilon_l E_{VY} - \alpha E_{VY}$$

$$\frac{dI_Y}{dt} = \epsilon_Y E_Y + yI_F + \alpha I_{VY} - (d_Y + h_Y)I_Y - aI_Y$$

$$\frac{dI_{VY}}{dt} = \epsilon_l \epsilon_Y E_{VY} - (d_Y + h_Y + \alpha + a)I_{VY}$$

$$\frac{dV_Y}{dt} = -(d_Y + h_Y + \alpha + a)V_Y - \beta V_Y(I + \kappa(I_{VY} + I_{VA})) - \frac{\gamma V_Y P}{\tau + P}$$

Pulse vaccination in yearlings:

$$V_Y(t_V) = p(S_Y(t_V^-) + E_Y(t_V^-) + E_{VY}(t_V^-) + V_Y(t_V^-))$$

$$S_Y(t_V) = (1 - p)S_Y(t_V^-)$$

$$E_Y(t_V) = (1 - p)E_Y(t_V^-)$$

$$E_{VY}(t_V) = (1 - p)E_{VY}(t_V^-)$$

Adult Dynamics:

$$\frac{dS_A}{dt} = aS_Y - \beta S_A(I + \kappa(I_{VY} + I_{VA})) - \frac{\gamma S_A P}{\tau + P} - (d_A + h_A)S_A + \alpha V_A$$

$$\frac{dE_A}{dt} = \beta S_A(I + \kappa(I_{VY} + I_{VA})) + \frac{\gamma S_A P}{\tau + P} + aE_Y - (d_A + h_A)E_A - \epsilon_A E_A$$

$$\frac{dE_{VA}}{dt} = \beta E_{VA}(I + \kappa(I_{VY} + I_{VA})) + \frac{\gamma E_{VA} P}{\tau + P} + aE_{VY} - (d_A + \epsilon_A + h_A)E_{VA} - \epsilon_A \epsilon_l E_{VA} - \alpha E_{VA}$$

$$\frac{dI_A}{dt} = \epsilon_A E_A + aE_Y + \alpha I_{VA} - (d_A + h_A)I_A$$

$$\frac{dI_{VA}}{dt} = \epsilon_l \epsilon_A E_{VA} + aI_{VY} - (d_A + h_A + \alpha)I_{VA}$$

$$\frac{dV_A}{dt} = aV_Y - (d_A + h_A + \alpha)V_A - \beta V_A(I + \kappa(I_{VY} + I_{VA})) - \frac{\gamma V_A P}{\tau + P}$$

Pulse vaccination in adults:

$$V_A(t_V) = p(S_A(t_V^-) + E_A(t_V^-) + E_{VA}(t_V^-) + V_A(t_V^-))$$

$$S_A(t_V) = (1-p)S_A(t_V^-)$$

$$E_A(t_V) = (1-p)E_A(t_V^-)$$

$$E_{VA}(t_V) = (1-p)E_{VA}(t_V^-)$$

Environmental propagule dynamics:

$$\frac{dP}{dt} = \theta(I_F + I_Y + I_A + \kappa(I_{VY} + I_{VA})) - \sigma P \qquad (1)$$

## Parameter estimation and model analysis

To reduce the uncertainty due to nonidentifiable parameters, parameter identifiability analysis must be done before estimation. For this, we used the Identifiability Analysis package in Mathematica (v.13.1.0.0) [51]. Little is known about parameter values associated with bTB dynamics in deer. Since only bTB case numbers are available for model fitting, system 1 becomes unidentifiable. Thus, we fix some of the parameters (specifically, $\theta$, $\gamma$, and initial conditions for all variables). First, we set the vaccination level and all other vaccine parameters to 0. Then, we estimated parameters using approximate Bayesian computation [52] assuming uniform priors and fitting the reduced model to bTB surveillance data obtained from Michigan Department of Natural Resources (MDNR) from years 2011 to 2021. The ranges of priors used for estimation are given in Table 1. We used Latin Hypercube sampling of parameter space using *lhs* package in R for generating prior parameter sets [53]. We simulated the model for 20 years (burn-in period) and used the remainder outputs from model simulation timepoints for parameter estimation. This process was repeated for each county associated with DMU 452. Point estimation and credible intervals of disease prevalence in each county were calculated from model simulations with random sampling of posterior parameters sets. The workflow for this part of the model is shown in S1 Fig.

We considered two strategies to reduce or eliminate bTB in white-tailed deer. First, we increased the harvest rates with the reduced model where vaccination is set to 0. We chose multiples of 3 years of management to match the review cycle from MDNR [Chad Stewart, personal communication]. Second, we examined yearly pulse vaccinations using a wide range of vaccine coverage (30% to 90% of susceptible and exposed deer populations). We also combined pulse vaccination and increased harvest to evaluate integrated management strategies. To understand how vaccine properties affected prevalence reduction, we used boosted regression trees to rank the influence of vaccine parameters on prevalence reduction when vaccine was applied annually for 30 years. To do this, we used the "GradientBoostedTrees" method in Mathematica (v.13.1.0.0) and ranked vaccine parameters: vaccine coverage, leakiness, relative transmissibility, and immunity waning rate. First, we obtained a parametric solution to system 1 in terms of the four parameters. Initial prevalence was taken just before the start of vaccination. Then maximum prevalence reduction was obtained by subtracting the minimum prevalence obtained during the vaccination period from the initial prevalence using wide range of parameter values (i.e., $p \in [0.3, 0.9]$, $\epsilon_L \in [0,1]$, $\kappa \in [0,1]$, and $\alpha \in [0.01, 0.07]$). The table containing maximum prevalence reduction and their corresponding parameter values were fed into the gradient boosted algorithm. SHapely Additive exPlanation (SHAP) framework was used to assign importance of each vaccine parameter for a particular prediction [54]. Average feature

**Table 1. Parameter definitions and range of values associated with bovine tuberculosis (bTB) in white-tailed deer considered for parameter estimation and simulation of the model.** The prior ranges are specific to Michigan, USA and are informed from previous studies [38].

| Parameter | Definition | Prior range/Values (unit) |
|---|---|---|
| $b_y$ | Per capita birth rate due to yearling | 0.5–0.9 (/year) |
| $b_a$ | Per capita birth rate due to adult | 0.5–1.2 (/year) |
| $\beta$ | Rate of exposure from direct contact with infected individual | $10^{-4}$–$3.5*10^{-4}$ (/year) |
| $\gamma$ | Rate of exposure from environmental propagules | $10^{-8}$ (/year) |
| $\kappa$ | Ratio of transmission rate from vaccinated individual to unvaccinated individual (a.k.a., relative transmissibility) | 0–1 |
| $d_F$ | Per capita natural death rate of fawn | 0.1–0.25 (/year) |
| $h_F$ | Per capita harvest rate of fawn | 0.1 (/year) |
| $\epsilon_F$ | Per capita rate of transition from exposed to infected in fawn | 0.01–0.05 (/year) |
| $y$ | Per capita rate of aging from fawn to yearling | 1 |
| $K$ | Carrying capacity for deer | 50,000–100,000 |
| $d_Y$ | Per capita natural death rate of yearling | 0.1–0.3 (/year) |
| $h_Y$ | Per capita harvest rate of yearling | 0.15 (/year) |
| $\epsilon_Y$ | Per capita rate of transition from exposed to infected in yearling | 0.01–0.05 (/year) |
| $a$ | Per capita rate of aging from yearling to adult | 1 |
| $p$ | Vaccine coverage of the population (proportion) | 0.3–0.9 |
| $\alpha$ | Per capita immune waning rate | 0.24–0.72 (/year) |
| $d_A$ | Per capita natural death rate of adult | 0.1–0.3 (/year) |
| $h_A$ | Per capita harvest rate of adult | 0.2 (/year) |
| $\epsilon_A$ | Per capita rate of transition from exposed to infected in adult | 0.01–0.05 (/ year) |
| $\theta$ | Per capita shedding rate of environmental propagule | 1300 (/year) |
| $\sigma$ | Decay rate of environmental propagule | 3–12 (/year) |

impact of the parameter values was then estimated as the mean absolute of Shapely values and the vaccine parameters were ranked accordingly. Lastly, we evaluated how these vaccine parameters influence time to elimination at county level.

## Results

### Disease dynamics in deer populations

The model shows county level differences in the disease prevalence trends in adults (Fig 2). Prevalence increased in Alpena and Alcona counties over the ten-year period, and decreased in Oscoda County and in Montmorency County. Because we used a deterministic model, it did not capture year to year variation in prevalence, and the trends are trajectories from the model output rather than statistical trends. Linear fit of apparent prevalence shows similar trends (S21 Fig). Given the large number (n = 11) of parameters estimated using only apparent prevalence data, a wide range of parameter combinations can show a good fit to the data. Sensitivity analysis shows that disease prevalence is more sensitive to environmental exposure rate compared to direct exposure rate to infected individual (S18 Fig). Although there were differences in the estimated median parameter values for each county, there were no clear patterns in the parameter values that explained differences in the trajectories of prevalence in each county (S2–S5 Figs). For example, even though the estimated median direct transmission rate is slightly higher in Montmorency and Oscoda counties, the trend in disease prevalence is decreasing. The number of deer estimated in these counties are lower compared to Alpena and Alcona (S17 Fig) that are likely driving these trends.

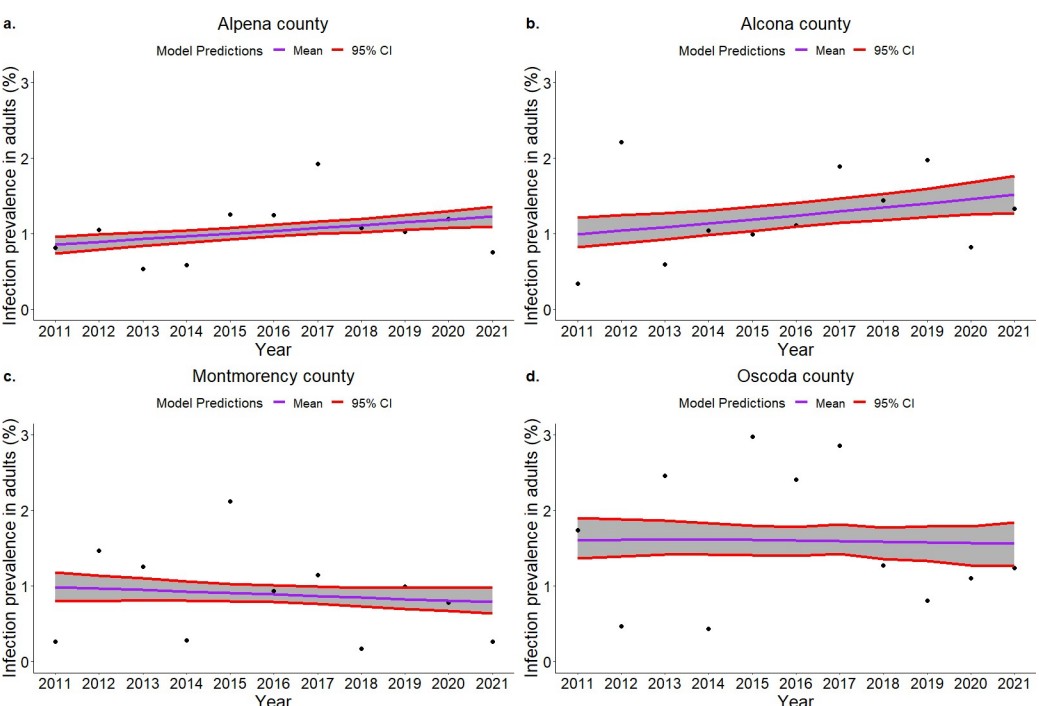

**Fig 2.** Model fit to adult white-tailed deer apparent disease prevalence in a. Alpena, b. Alcona c. Montmorency, and d. Oscoda counties. Purple line shows model predictions from estimated median parameters for each county and shaded region between red lines represent 95% credible interval obtained from 1000 simulations.

### Harvest as a disease control measure

Sensitivity analysis demonstrated that disease prevalence was most sensitive to adult harvest rate (S6 and S7 Figs). Similar sensitivity to adult harvest rate was observed when pulse harvest was considered (S20 Fig). Hence, we modified adult harvest rate to evaluate its influence in reducing disease prevalence in each county (Fig 3A). Prevalence was most affected by

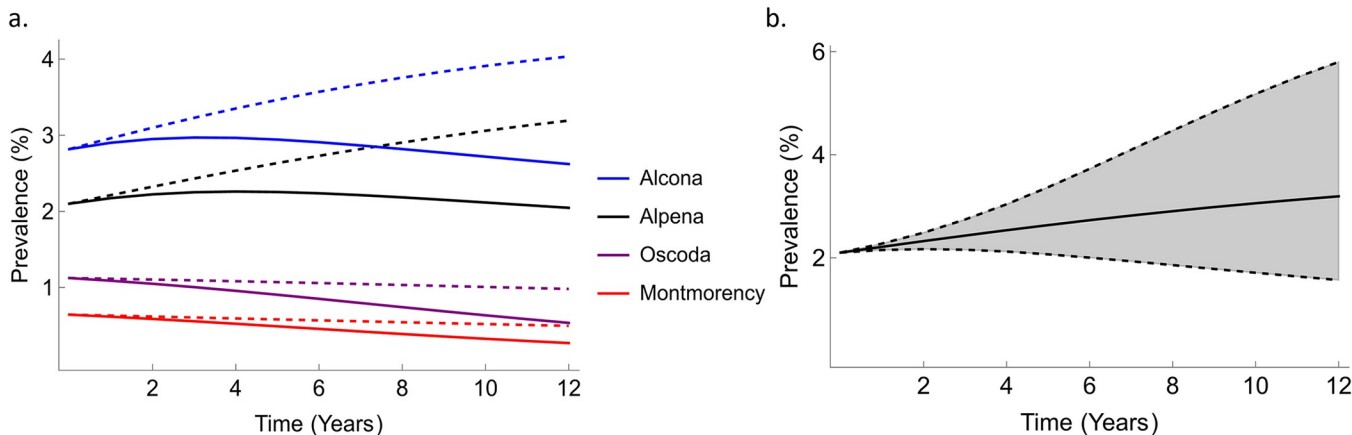

**Fig 3.** a. Effects of increased adult harvest rates on bTB prevalence in four counties. Dashed lines show baseline projections of infection prevalence whereas solid lines represent projections when adult harvest rate is increased by 30%. Estimated median parameter values are used for each county. b. Range of bTB prevalence changes for 50% changes in harvest rates in Alpena County. The upper dashed line represents results for a 50% decrease in harvest rate, lower represents the result for a 50% increase in harvest rate and the solid line represents baseline predictions (maintaining harvest at current levels).

increased (30%) harvest in Alpena (reduced by 1.19% points) and Alcona (reduced by 1.41% points) counties, whereas very little effect was seen in Montmorency (reduced by 0.23% points) and Oscoda counties (reduced by 0.45% points). In Alpena County a 50% decrease in current adult harvest rates led to a 100% increase in disease prevalence within the next 12 years (Fig 3B).

## Pulse vaccination as a disease control measure

Even low vaccination coverage significantly reduced disease prevalence within 6 years compared to a no-vaccination scenario (Figs 4A–4D; black dashed dot). For example, 30% coverage with an 80% efficacious vaccine reduced disease prevalence by 40% in Alpena County within 8 years (Fig 4A). Long-term projection of prevalence shows that the short-term vaccination efforts can have long lasting effects on prevalence trends (S16 Fig). The disease prevalence was reduced by 70% if the annual vaccination was continued for 15 years (S8 Fig). There were diminishing returns of vaccine coverage—medium coverage was sufficient in greatly reducing infection prevalence (red continuous and red-dashed lines in Fig 4), which was especially pronounced for long-term (15 years) vaccination (S8 Fig). Reducing vaccination frequency to every 2 years led to lesser and slower reduction in disease prevalence compared to an annual vaccination regime. However, the difference was small for high vaccine coverage and a longer vaccination schedule suggesting that, in the long run, similar reduction in disease prevalence was observed with biennial vaccination (S8 Fig). Similar results were observed for vaccinating every 3 years (S14 Fig).

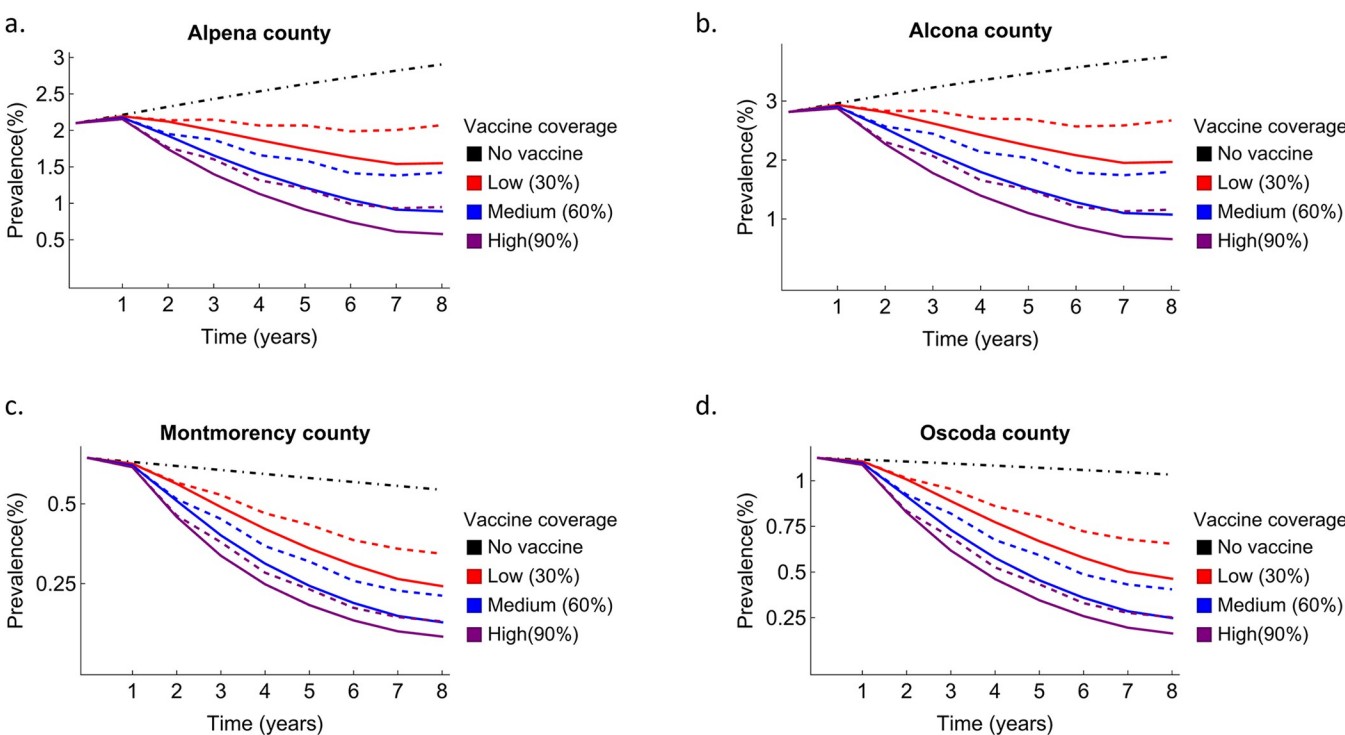

**Fig 4.** Short-term vaccination effect on bTB prevalence in white-tailed deer in a. Alpena County, b. Alcona County, c. Montmorency County and d. Oscoda County in Michigan, USA. Vaccination was applied from year 1 to year 6. Black dashed and dotted lines represent baseline scenarios with no vaccination. Continuous lines show prevalence for yearly vaccination at different vaccination coverage. Dashed lines show prevalence assuming biennial vaccination for the same time. All simulations for each county were done using median parameter estimates. Other vaccine parameter values included $\epsilon_L = 0.2$, $\kappa = 0.5$, and $\alpha = 0.06$.

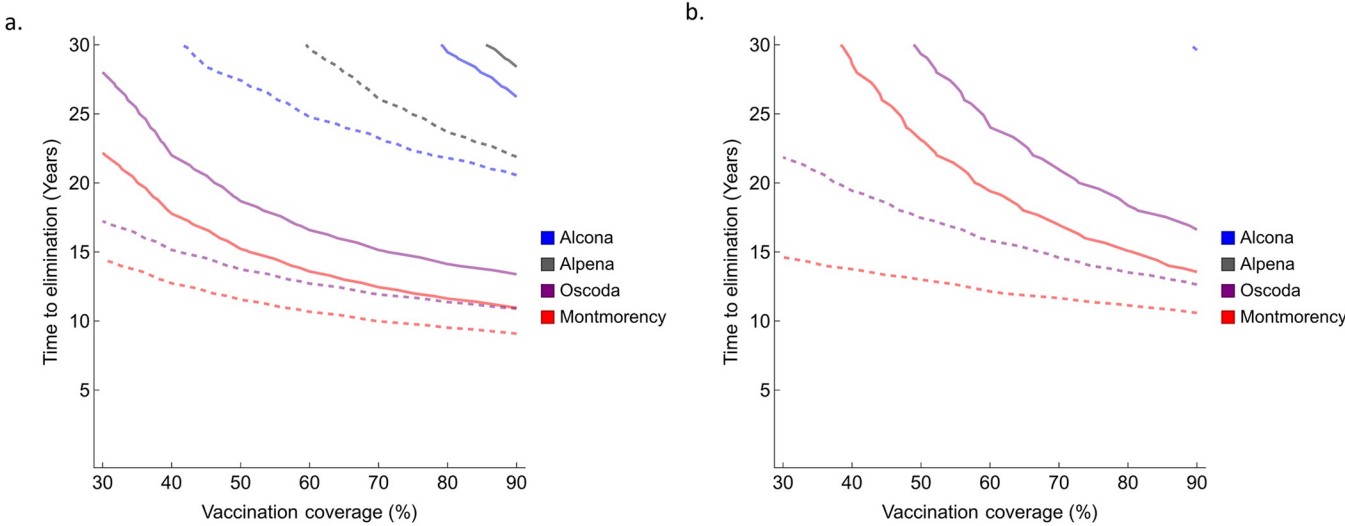

**Fig 5. Time to disease elimination based on different vaccination and harvest levels in four counties in Michigan, USA.** Gray, blue, purple, and red represent Alpena, Alcona, Oscoda, and Montmorency counties, respectively. a. shows annual vaccination whereas b. shows biennial vaccination. Continuous lines represent time to elimination with current levels of adult harvest whereas dashed lines represent the time to elimination when harvest is increased by 30% of the current values. All estimates are based on median parameter values estimated for each county. Other vaccine parameter values include $\epsilon_L = 0.2$, $\kappa = 0.5$, and $\alpha = 0.06$. Lines for Alcona and Alpena counties are missing in b. because elimination did not occur within 30 years.

## Combining harvest and pulse vaccination for disease elimination

When vaccination of deer was the only management intervention, more than 80% vaccination coverage was required to reach bTB elimination in Alpena and Alcona Counties within 30 years (Fig 5A, blue and gray solid lines). But, if the adult harvest rate was also increased by 30%, time to elimination reduced significantly (~20% reduction at 90% vaccine coverage) and could be achieved with moderate vaccination coverage (Fig 5A, blue and gray dashed lines). This is due to the reduced deer number with increased harvest rate (S17 Fig). However, disease elimination in these two counties was unachievable under biennial vaccination for 30 years (Fig 5B). In Oscoda and Montmorency counties, where the disease prevalence trend is decreasing (Fig 2), disease elimination could be achieved using only vaccination even with moderate vaccine coverage. Elimination of bTB in these counties could be achieved within a 30-year time frame even with a biennial vaccination regime. Time to elimination can further be reduced by increasing adult harvest rates (Fig 5, purple and red dashed lines).

## Vaccine properties in prevalence reduction and disease elimination

We used gradient boosted regression to evaluate the relative importance of four vaccine parameters in reducing disease prevalence. A SHapely additive explanations framework was used to sort the relative influence of each parameter in prevalence reductions in the four counties. Under an annual vaccination regime, vaccine coverage was the most important feature in prevalence reduction followed by leakiness and immunity waning rate in all four counties (Figs 6A–6D), although the magnitude of each feature differed among counties. This pattern was robust to vaccination frequency (e.g., S13 Fig). Relative transmissibility had the smallest effect on prevalence reduction. Time to elimination was most sensitive to vaccine coverage and leakiness under an annual vaccination regime, whereas relative transmissibility and immune waning rate had little influence (S9–S12 Figs).

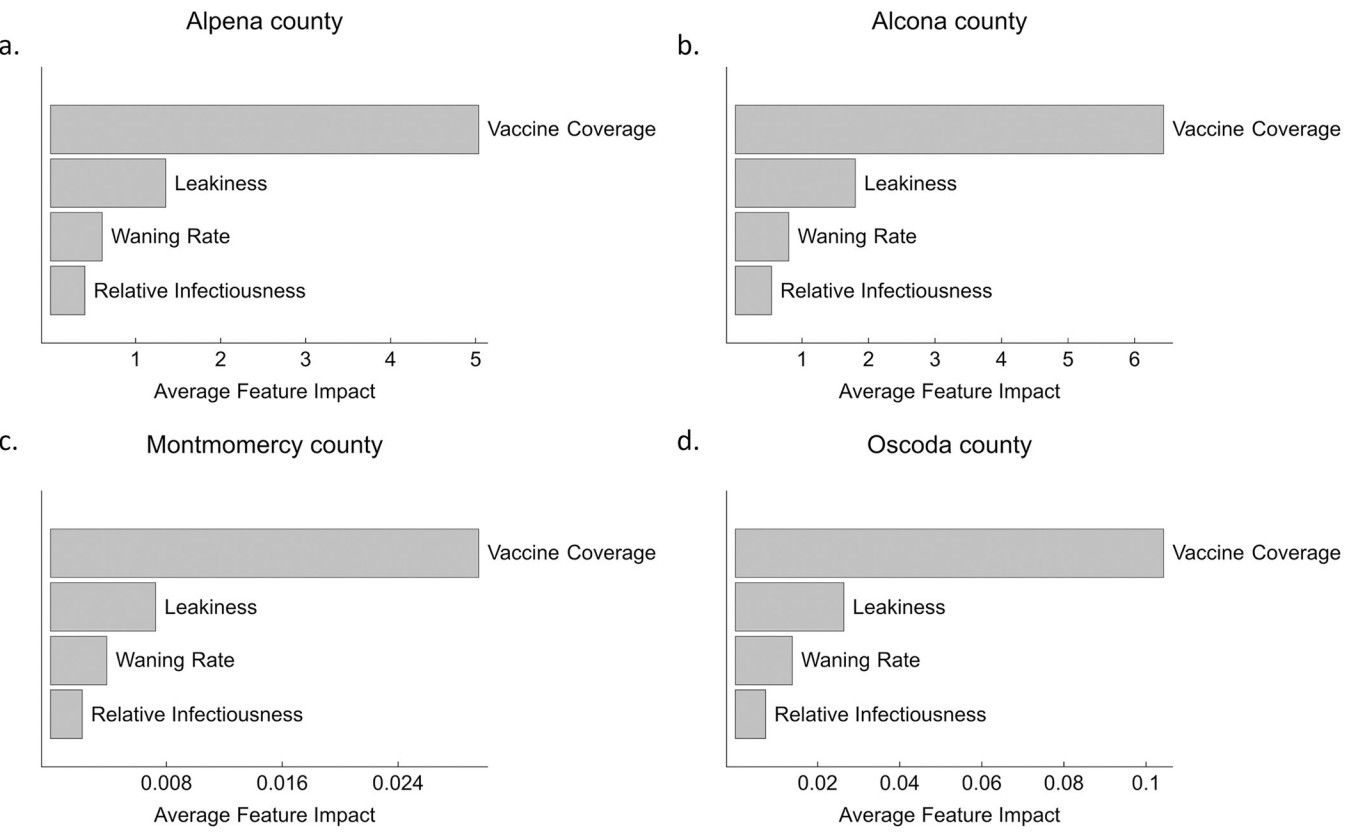

**Fig 6.** Average feature impact of vaccine properties for yearly vaccination based on Shapely additive explanation values obtained for prevalence reduction in a. Alpena, b. Alcona, c. Montmorency, and d. Oscoda counties. Estimated median parameter values for each county is used. See methods section for the range of vaccine properties used.

## Discussion

Fitting a disease transmission model with surveillance data from Michigan, we identified integrated management strategies involving vaccination and harvest that effectively managed, and even eliminated, bTB from local white-tailed deer populations. Increasing harvest beyond current levels had the greatest effect in the counties where disease prevalence has increasing trend (Alpena and Alcona), whereas vaccination could be used solely to eliminate bTB in counties where prevalence is decreasing (Oscoda and Montmorency). Additional harvest rate further contributes to prevalence reduction in Oscoda and Montmorency reducing time to elimination. We chose to evaluate the changes to bTB prevalence through changes to adult harvest, because it is the age class containing the largest share of positive cases. Our model included a mix of direct and environmental transmission, thus including both density-dependent and frequency-dependent transmission processes [55]. The inclusion of frequency-dependent transmission can reduce the potential for harvest to reduce disease transmission that works by reducing deer density and thus direct transmission (a density-dependent function). In this case, disease reduction depends on the relative strength of the two modes of transmission. Regardless, harvesting can still be an effective strategy even in some frequency-dependent systems, especially when a population shows density dependent birth (as assumed in our model) [56].

Given the potential ethical, ecological, and economical concerns with culling or increased harvest in wildlife host species, vaccination is proposed either as a supplement or alternative

technique for managing disease in wildlife. Our fitted model predicted that even a moderate vaccine coverage (60%) with a vaccine that is 80% effective can greatly reduce disease prevalence within six years. Previous research with biomarker trials has shown that up to 69% vaccination coverage can be achieved with vaccine delivery units for white-tailed deer in this area [49]. Our work suggests that this uptake rate is more than sufficient to achieve large reductions (>90% within 15 years) in bTB prevalence, and that increased vaccination coverage would not dramatically increase the effect, which is helpful in optimizing vaccination efforts under economic and potentially social constraints. A previous model of partial vaccine protection in badgers found that long term vaccination with imperfect vaccines can eliminate disease from local a population [57]. Consistent with this, we show county-wide elimination with long-term annual vaccinations where time to elimination can be reduced when vaccination is combined with increased harvest as shown in other studies [39]. Although annual vaccination led to the maximum reduction, biennial or triennial vaccination still led to a significant reduction in disease prevalence, which could be enhanced with increased coverage. Hence, our work shows that wildlife vaccination could significantly reduce prevalence and lays out a framework for designing optimal vaccination schedules.

The extent and duration of protection offered by the BCG vaccine in wild deer populations is still an open question. Research in cattle [50] and badgers [58] show partial protection from BCG vaccine, a likely scenario in deer [32]. BCG vaccine has been shown to reduce clinical signs rather than providing complete protection and its protection wanes with time [59]. However, most models of vaccination assume perfect and/or lifetime immune protection [60]. Vaccine coverage followed by leakiness were the two most important parameters in prevalence reduction in our system, while immunity waning rate and relative transmissibility had little influence. This is slightly different than the results from continuous vaccination models with imperfect vaccine against severe acute respiratory syndrome (SARS) and tuberculosis (*M. tuberculosis*) that showed both coverage and efficacy had equivalent impact in disease reduction [61,62]. New and potentially more effective BCG vaccines with differing levels of immune protection and coverages are being explored [36,63,64]. High vaccine coverage is more likely with transmissible vaccines (e.g., [65,66], and some preliminary results show potential for limited transmission of BCG between deer [33,67]. Thus, research focused on methods of achieving adequate vaccination coverage and vaccine development focused on reducing leakiness are likely to produce the best results for management.

As with any modeling study, the results presented are critically dependent upon model assumptions, some of which might not be realistic. There is a lack of detailed knowledge on how bTB progresses and transmits in free-ranging white-tailed deer populations making it difficult to parametrize the system. Additionally, we have assumed that there is no difference in disease dynamics between sexes which may not be true [45]. Further, preferential harvest of adult female can lead to greater reduction in deer population and thus our results may underestimate the reduction in disease prevalence because of increased harvest rate. Spatial heterogeneity and animal movement patterns can also influence disease transmission and the effectiveness of management including vaccination [68], which our model does not addresses. Similarly, the model is fitted to apparent prevalence which itself can be biased. At this time, the samples submitted that contain gross lesions are the only ones that are considered for further testing and samples from different surveillance methods have different likelihoods of finding bTB-positive deer [69], likely leading to underreporting of cases [70,71]. Underreporting can influence the estimated parameter values and ultimately the management recommendations made here. For example, a preliminary analysis with assumed higher disease prevalence in Alpena County shows a similar trend in prevalence reduction to what we have shown here with short-term vaccination (S19 Fig), however, this changes the timeframe to disease

elimination. Additionally, *M. bovis* has a broad host range and interactions between these hosts can change the disease dynamics in deer and influence the outcomes of interventions [72]. Thus, a better understanding of disease dynamics in free-ranging deer populations is needed for robust management recommendations.

Models of wildlife disease dynamics with potential interventions can provide useful information for evaluating and guiding bTB management strategies [5,60,73]. Although the general principles of disease ecology can be applied across systems, system and locality specific models are often the best tools for informing management decisions. By fitting the model at the county level, we show how the same management practices can have differing outcomes in counties with differing disease prevalence. At the same time, we provide a basis for using harvest and pulse vaccination in endemic wildlife disease management of a chronic disease. By highlighting the important properties of vaccines epidemiologically, we provide guidance to further vaccine research for effective wildlife disease management. Thus, integrating disease ecology principles using models prior to deploying management campaigns can provide useful guidance for both vaccine product development and field-based deployment strategies.

## Supporting information

**S1 Fig. Model workflow for estimating county-level model parameters and resulting model outputs associated with model parameters.**
(TIF)

**S2 Fig. Distribution of estimated parameter values for Alpena County Michigan. Distribution shows 1000 estimated parameter values using approximate Bayesian computation. Dashed vertical lines show median values from the distribution.**
(TIF)

**S3 Fig. Distribution of estimated parameter values for Alcona County Michigan.** Distribution shows 1000 estimated parameter values using approximate Bayesian computation. Dashed vertical lines show median values from the distribution.
(TIF)

**S4 Fig. Distribution of estimated parameter values for Montmorency County Michigan.** Distribution shows 1000 estimated parameter values using approximate Bayesian computation. Dashed vertical lines show median values from the distribution.
(TIF)

**S5 Fig. Distribution of estimated parameter values for Oscoda County Michigan.** Distribution shows 1000 estimated parameter values using approximate Bayesian computation. Dashed vertical lines show median values from the distribution.
(TIF)

**S6 Fig. Sensitivity of adult and yearling harvest rates on disease prevalence.** All other parameters are median estimated parameters for Alpena County Michigan assuming no vaccination. See Table 1 for the parameters used.
(TIF)

**S7 Fig. Changes in disease prevalence in Alpena County Michigan estimated for the next 12 years based on 50% changes on the current yearling white-tailed deer harvest rates.** Upper dashed line shows scenario when adult deer harvest rate is decreased by 50% and lower dashed line shows scenario when the adult harvest rate is increased by 50%. Solid line represents baseline projection with no change in harvest rate. All other parameters used for the Figs

are estimated median parameter values for Alpena County.
(TIF)

**S8 Fig. Long-term vaccination effect on disease prevalence in white-tailed deer in 4 counties.** Vaccination was applied from year 1 to year 16. Black dashed and dotted line represent baseline scenarios with no vaccination. Continuous lines show prevalence for yearly vaccination at different vaccine coverage. Dashed lines show prevalence assuming biennial vaccination for the same time. All simulations for each county were done using median estimated parameter values.
(TIF)

**S9 Fig. Time to disease elimination based on different vaccine coverage and white-tailed deer harvest levels in four counties.** Gray, blue, purple, and red represent Alpena, Alcona, Oscoda, and Montmorency counties in Michigan, respectively. Continuous lines represent time of elimination with current level of adult harvest whereas dashed lines represent the time of elimination when harvest is increased by 30% of the current values. Other vaccine parameters include $\epsilon_L = 0.2$, $\kappa = 0.5$, and $\alpha = 0.06$.
(TIF)

**S10 Fig. Time to disease elimination based on vaccine leakiness and different harvest levels in four counties.** Gray, blue, purple, and red represent Alpena, Alcona, Oscoda, and Montmorency counties, respectively. Continuous lines represent time of elimination with current level of adult harvest whereas dashed lines represent the time of elimination when harvest is increased by 30% of the current values. Other vaccine parameters include $p = 0.7$, $\kappa = 0.5$, and $\alpha = 0.06$.
(TIF)

**S11 Fig. Time to disease elimination based on relative infectiousness and different white-tailed deer harvest levels in four counties.** Black, blue, purple, and red represent Alpena, Alcona, Oscoda, and Montmorency counties, respectively. Continuous lines represent time of elimination with current level of adult harvest whereas dashed lines represent the time of elimination when harvest is increased by 30% of the current values. Other vaccine parameters include $\epsilon_L = 0.2$, $p = 0.7$, and $\alpha = 0.06$.
(TIF)

**S12 Fig. Time to disease elimination based on immune waning rate and different harvest levels of white-tailed deer in four counties.** Black, blue, purple, and red represent Alpena, Alcona, Oscoda, and Montmorency counties, respectively. Continuous lines represent time of elimination with current level of adult harvest whereas dashed lines represent the time of elimination when harvest is increased by 30% of the current values. Other vaccine parameters include $\epsilon_L = 0.2$, $p = 0.7$, and $\kappa = 0.5$.
(TIF)

**S13 Fig. Average feature impact of vaccine parameters for the vaccination of white-tailed deer every 3 years based on Shapely additive explanation values obtained for prevalence reduction in each county.** Estimated median parameter values for each county is used. See method section for the range of vaccine parameters used to obtain the results.
(TIF)

**S14 Fig. Long-term vaccination of white-tailed deer effect on disease prevalence in 4 counties.** Vaccination was applied from year 1 to year 16. Black dashed and dotted line represent baseline scenarios with no vaccination. Continuous lines show prevalence for yearly

vaccination at different vaccine coverage. Dashed lines show prevalence assuming vaccination every 3 years for the same time frame. All simulations for each county are done using median estimated parameter values.
(TIF)

**S15 Fig. Deer dynamics in Alpena County.** The red line shows the number of adult deer, blue line represents number of yearling deer and green shows number of fawns. The fluctuation is due to birth pulses. The deer number estimates are obtained from the main model using median estimated parameter values.
(TIF)

**S16 Fig. Long term projection for the effect of short-term vaccination on bTB prevalence in white-tailed deer in four counties in Michigan, USA.** Vaccination was applied from year 1 to year 6. Black dashed and dotted lines represent baseline scenarios with no vaccination. Continuous lines show prevalence for yearly vaccination at different vaccination coverage. Dashed lines show prevalence assuming biennial vaccination for the same time. All simulations for each county were done using median parameter estimates. Other vaccine parameter values included $\epsilon_L = 0.2$, $\kappa = 0.5$, and $\alpha = 0.06$.
(TIF)

**S17 Fig. Effect of vaccination and harvest on deer population sizes in four counties.** Continuous lines represent deer population with current level of harvest rates whereas dashed lines represent deer population when the harvest rate is increased by 30%. We assumed vaccine coverage to be 70% and yearly vaccination for 30 years. All other parameters are the same as in Fig 5A.
(TIF)

**S18 Fig. Sensitivity of endemic disease prevalence on direct and environmental exposure rates.** All other parameters are estimated median parameters for Alpena County.
(TIF)

**S19 Fig. Short-term vaccination effect on assumed higher bTB prevalence in white-tailed deer in Alpena County in Michigan, USA.** The parameters were estimated assuming 5X that the 'true prevalence' is 5X the observed apparent prevalence. Vaccination was applied from year 1 to year 6. Black dashed and dotted lines represent baseline scenarios with no vaccination. Continuous lines show prevalence for yearly vaccination at different vaccination coverage. Dashed lines show prevalence assuming biennial vaccination for the same time. All simulations for each county were done using median parameter estimates. Other vaccine parameter values included $\epsilon_L = 0.2$, $\kappa = 0.5$, and $\alpha = 0.06$.
(TIF)

**S20 Fig. Sensitivity of adult and yearling harvest rates on disease prevalence with harvest being considered as a pulse.** For this, we assume that a proportion of a given age class of deer are harvested during a month. More specifically, we assume it occurs during December every year. Transmission rates are adjusted to fit the data to the modified model. All other parameters are median estimated parameters for Alpena County Michigan assuming no vaccination. See Table 1 for the parameters used.
(TIF)

**S21 Fig. Linear regression of infection prevalence in adults in Alpena, Alcona, Montmorency, and Oscoda counties.**
(TIF)

**S1 Table. Infection prevalence data by age and year in Alpena County.**
(XLSX)

**S2 Table. Infection prevalence data by age and year in Alcona County.**
(XLSX)

**S3 Table. Infection prevalence data by age and year in Montmorency County.**
(XLSX)

**S4 Table. Infection prevalence data by age and year in Oscoda County.**
(XLSX)

## Acknowledgments

We thank the Michigan Department of Natural Resources-Wildlife Division for providing data critical for the analysis of the model. Insights and resources from Dr. Mitch Palmer (USDA, ARS) were helpful in developing the vaccination model.

## Author Contributions

**Conceptualization:** Kurt C. VerCauteren, Henry Campa, III, Kim M. Pepin.

**Data curation:** Abigail B. Feuka, Melinda Cosgrove, Megan Moriarty, Anthony Duffiney.

**Formal analysis:** Aakash Pandey, Abigail B. Feuka.

**Funding acquisition:** Kurt C. VerCauteren, Henry Campa, III, Kim M. Pepin.

**Investigation:** Aakash Pandey.

**Methodology:** Aakash Pandey, Kim M. Pepin.

**Resources:** Henry Campa, III.

**Supervision:** Kurt C. VerCauteren, Henry Campa, III, Kim M. Pepin.

**Visualization:** Aakash Pandey.

**Writing – original draft:** Aakash Pandey.

**Writing – review & editing:** Aakash Pandey, Abigail B. Feuka, Melinda Cosgrove, Megan Moriarty, Anthony Duffiney, Kurt C. VerCauteren, Henry Campa, III, Kim M. Pepin.

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
