## [Decision Letter · Decision Letter 0]

31 Jul 2023

Dear Dr. Pandey,

Thank you very much for submitting your manuscript "Wildlife vaccination strategies for eliminating bovine tuberculosis at the wildlife-livestock interface" for consideration at PLOS Computational Biology.

As with all papers reviewed by the journal, your manuscript was reviewed by members of the editorial board and by several independent reviewers. In light of the reviews (below this email), we would like to invite the resubmission of a significantly-revised version that takes into account the reviewers' comments.

We cannot make any decision about publication until we have seen the revised manuscript and your response to the reviewers' comments. Your revised manuscript is also likely to be sent to reviewers for further evaluation.

Sincerely,

Yamir Moreno

Academic Editor

PLOS Computational Biology

Virginia Pitzer

Section Editor

PLOS Computational Biology

Reviewer's Responses to Questions

**Comments to the Authors:**

Reviewer #1: Pandey et al. have used a modeling approach to assess vaccination strategies for eliminating bTB in Michigan white-tailed deer populations. This topic is very relevant to managing a serious wildlife health, public health and economic issue, and timely. I have the following suggestions/comments:

1) bTB surely is an excellent example of a pathogen transmitted at the wildlife-domestic-human interface, but your work exclusively focuses on the wildlife component - you have evaluated strategies for bTB elimination /control in WTD populations, right? I suggest replacing the 'wildlife-livestock interface' in your title to 'white-tailed deer populations'.

2) You compare model results for 4 counties - yet there is no countywise deer population data provided in your paper. How does the N differ between these counties? How does the age composition differ between these counties? Did you use pre-harvest or post-harvest estimates?

3) You have provided parameter values in Table 1. All except the per capita immune waning rate are yearly (annual) rates. How do you account for this in your model?

4) Moreover, you have assumed that per capita harvest rates are same for male and female deer. Could you provide some explanation why you made this assumption when the harvest rates in the real world are clearly different in male and female deer? Do consider that bTB prevalence differs significantly between males (8%) and females (2%) as documented by O'Brien et al 2002. I might have missed any current references on this topic; if so, please include these references.

5) This also brings up the question about the sensitivity analysis - you state that disease prevalence is very sensitive to adult harvest rates. But you make unrealistic assumption about male and female harvest rates. How valid are your conclusions in the context of this assumption?

6) Line 290: Should'nt it be Figure 2?

7) Another critical component missing from your results is the impact of these scenarios on the deer population. For scenarios where harvest and vaccination are combined, could you provide a plot to show the population size of deer? Would help the reader understand the context for these results.

8) Figure 4 would provide critical insights if you extend the x axis (years) to 30? How long does the effect of short term vaccination last?

Reviewer #2: The authors develop an age structured mechanistic model that is fitted on bovine TB surveillance data in white tailed deer in Michigan. They use the models to test effect vaccines (different efficacy and coverage) and deer harvesting on bTB transmission dynamics. The manuscript is beautifully written with a good flow of logic in the introduction, and good explanation of the methods as well as good presentation of the results, and insights drawn from the model. I enjoyed reading it.

I have two main comments for consideration by the authors:

- What is relative contribution of live propagules in the environment to the main outcome modelled (prevalence of bTB)? This number of live propagules in the environment would be expected to be a function of prevalence of bTB in the study population - how is this accounted for in the model?

- The authors rightfully identify sampling bias as a potential limitation to the data used to parametarize the model. I would be keen to see how the model results and interpretation of insights would vary if under ascertainment of the infection rates was incorporated in the model.

**Have the authors made all data and (if applicable) computational code underlying the findings in their manuscript fully available?**

Reviewer #1: Yes

Reviewer #2: Yes

PLOS authors have the option to publish the peer review history of their article (what does this mean?). If published, this will include your full peer review and any attached files.

Reviewer #1: **Yes: **Aniruddha Belsare

Reviewer #2: No
---

## [Decision Letter · Decision Letter 1]

20 Nov 2023

Dear Dr. Pandey,

Thank you very much for submitting your manuscript "Wildlife vaccination strategies for eliminating bovine tuberculosis in white-tailed deer populations" for consideration at PLOS Computational Biology.

As with all papers reviewed by the journal, your manuscript was reviewed by members of the editorial board and by an independent reviewer. In light of the review (below this email), we would like to invite the resubmission of a significantly-revised version that takes into account the reviewer's comments.

Please carefully address the critical comment 1 of the reviewer, which might lead to methodological problems. It may also be helpful to implement a sensitivity analysis testing for the robustness of such an assumption/procedure.

Sincerely,

Yamir Moreno

Academic Editor

PLOS Computational Biology

Virginia Pitzer

Section Editor

PLOS Computational Biology

Reviewer's Responses to Questions

**Comments to the Authors:**

Reviewer #1: Thank you for responding to my comments. I have a few follow-up questions:

1) You state in your response to Comment 2: "In the model, we have assumed constant/continuous harvest of deer throughout the year for simplification. This is further supported by the fact that the USDA removes deer outside of the hunting season." I think this simplistic assumption is problematic. Just considering the harvest data for Alpena County, about 10% of the deer (~2500) are removed (harvested) in the month of November; a total of ~15% (3500) removed during the 4 months of harvest season (Sep-Dec). But, as per your explanation, you are simply dividing the annual harvest rate by 12 to get a monthly harvest rate, so you have 400 deer harvested every month in your model. The sensitivity analysis (Fig. S6) is questionable due to the unrealistic monthly harvest rates. Such simplifying assumptions influence model dynamics and disease dynamics, and in my opinion, negatively impact the applicability and utility of your modeling approach.

2) Do consider converting the finite rates to instantaneous rates instead of just dividing annual rates by 12. See https://influentialpoints.com/Training/finite-and-instantaneous_rates.htm

3) You state that (Line 305) "Prevalence increased in Alpena and Alcona counties over the ten-year period, remained level in Oscoda County and decreased in Montmorency County." Is this based on 'eyeballing' Figure 2, or you used some other statistic? Please provide details here.

4) Similarly, you state (Line 324) "Prevalence was most affected by increase (30%) harvest in Alpena and Alcona counties, whereas very little effect was seen in Montmorency and Oscoda counties." How was this ('most affected', 'little effect') determined? Did you do statistical comparisons? If so, please provide details.

5) It is surprising that you did not discuss in more detail the decreasing prevalence of bTB in Oscoda and Montmorency counties (Figures 3, 4, S16, etc.). As you are using a compartmental model, this means the R0 is less than 1, but in the context of Fig. S17, it might just be due the population threshold size (Nt). This might not reflect real-world process, and majorly affects the relevance of your findings.

**Have the authors made all data and (if applicable) computational code underlying the findings in their manuscript fully available?**

Reviewer #1: Yes

PLOS authors have the option to publish the peer review history of their article (what does this mean?). If published, this will include your full peer review and any attached files.

Reviewer #1: **Yes: **Aniruddha Belsare
---

## [Decision Letter · Decision Letter 2]

12 Dec 2023

Dear Dr. Pandey,

We are pleased to inform you that your manuscript 'Wildlife vaccination strategies for eliminating bovine tuberculosis in white-tailed deer populations' has been provisionally accepted for publication in PLOS Computational Biology.

Best regards,

Yamir Moreno

Academic Editor

PLOS Computational Biology

Virginia Pitzer

Section Editor

PLOS Computational Biology

Reviewer's Responses to Questions

**Comments to the Authors:**

Reviewer #1: Thanks for responding to my comments. Great job!

**Have the authors made all data and (if applicable) computational code underlying the findings in their manuscript fully available?**

Reviewer #1: Yes

PLOS authors have the option to publish the peer review history of their article (what does this mean?). If published, this will include your full peer review and any attached files.

Reviewer #1: **Yes: **Aniruddha V. Belsare

---

## [Editor Report · Acceptance letter]

22 Dec 2023

PCOMPBIOL-D-23-00973R2 

Wildlife vaccination strategies for eliminating bovine tuberculosis in white-tailed deer populations

Dear Dr Pandey,

I am pleased to inform you that your manuscript has been formally accepted for publication in PLOS Computational Biology. Your manuscript is now with our production department and you will be notified of the publication date in due course.

With kind regards,

Anita Estes
